# Computational modelling of cell motility modes emerging from cell-matrix adhesion dynamics

Leonie van Steijn[1], Inge M. N. Wortel[2], Clément Sire[3], Loïc Dupré[4,5,6], Guy Theraulaz[7,8], Roeland M. H. Merks[1,9]*

1 Mathematical Institute, Leiden University, Leiden, The Netherlands, 2 Data Science, Institute for Computing and Information Sciences, Radboud University, Nijmegen, The Netherlands, 3 Laboratoire de Physique Théorique, Centre National de la Recherche Scientifique (CNRS) & Université de Toulouse—Paul Sabatier, Toulouse, France, 4 Toulouse Institute for Infectious and Inflammatory Diseases (INFINITy), INSERM, CNRS, Université de Toulouse, Toulouse, France, 5 Ludwig Boltzmann Institute for Rare and Undiagnosed Diseases, Vienna, Austria, 6 Department of Dermatology, Medical University of Vienna, Vienna, Austria, 7 Centre de Recherches sur la Cognition Animale (CRCA), Centre de Biologie Intégrative (CBI), Centre National de la Recherche Scientifique (CNRS) & Université de Toulouse—Paul Sabatier, Toulouse, France, 8 Centre for Ecological Sciences, Indian Institute of Science, Bengaluru, India, 9 Institute of Biology Leiden, Leiden University, Leiden, The Netherlands

* merksrmh@math.leidenuniv.nl

**Data Availability Statement:** All relevant data are within the manuscript and its Supporting information.

## Abstract

Lymphocytes have been described to perform different motility patterns such as Brownian random walks, persistent random walks, and Lévy walks. Depending on the conditions, such as confinement or the distribution of target cells, either Brownian or Lévy walks lead to more efficient interaction with the targets. The diversity of these motility patterns may be explained by an adaptive response to the surrounding extracellular matrix (ECM). Indeed, depending on the ECM composition, lymphocytes either display a floating motility without attaching to the ECM, or sliding and stepping motility with respectively continuous or discontinuous attachment to the ECM, or pivoting behaviour with sustained attachment to the ECM. Moreover, on the long term, lymphocytes either perform a persistent random walk or a Brownian-like movement depending on the ECM composition. How the ECM affects cell motility is still incompletely understood. Here, we integrate essential mechanistic details of the lymphocyte-matrix adhesions and lymphocyte intrinsic cytoskeletal induced cell propulsion into a Cellular Potts model (CPM). We show that the combination of *de novo* cell-matrix adhesion formation, adhesion growth and shrinkage, adhesion rupture, and feedback of adhesions onto cell propulsion recapitulates multiple lymphocyte behaviours, for different lymphocyte subsets and various substrates. With an increasing attachment area and increased adhesion strength, the cells' speed and persistence decreases. Additionally, the model predicts random walks with short-term persistent but long-term subdiffusive properties resulting in a pivoting type of motility. For small adhesion areas, the spatial distribution of adhesions emerges as a key factor influencing cell motility. Small adhesions at the front allow for more persistent motility than larger clusters at the back, despite a similar total adhesion area. In conclusion, we present an integrated framework to simulate the effects of ECM

**Funding:** This work was carried out on the Dutch national e-infrastructure with the support of SURF Cooperative (2019.001). L.v.S. was supported by the French Embassy in the Netherlands with the Bourse d'Excellence Descartes de stage. I.M.N.W. was funded by a Human Frontiers Science Program grant (HFSP, RGP0053/2020). R.M.H.M. was funded by the Nederlandse Organisatie voor Wetenschappelijk Onderzoek grant NWO/ENW-VICI 865.17.004. The funders had no role in study design, data collection and analysis, decision to publish, or preparation of the manuscript.

**Competing interests:** The authors have declared that no competing interests exist.

proteins on cell-matrix adhesion dynamics. The model reveals a sufficient set of principles explaining the plasticity of lymphocyte motility.

## Author summary

During immunosurveillance, lymphocytes patrol through tissues to interact with cancer cells, other immune cells, and pathogens. The efficiency of this process depends on the kinds of trajectories taken, ranging from simple Brownian walks to Lévy walks. The composition of the extracellular matrix (ECM), a network of macromolecules, affects the formation of cell-matrix adhesions, thus strongly influencing the way lymphocytes move. Here, we present a model of lymphocyte motility driven by adhesions that grow, shrink and rupture in response to the ECM and cellular forces. Compared to other models, our model is computationally light making it suitable for generating long term cell track data, while still capturing actin dynamics and adhesion turnover. Our model suggests that cell motility is affected by the force required to break adhesions and the rate at which new adhesions form. Adhesions can promote cell protrusion by inhibiting retrograde actin flow. After introducing this effect into the model, we found that it reduces the cellular diffusivity and that it promotes stick-slip behaviour. Furthermore, location and size of adhesion clusters determined cell persistence. Overall, our model explains the plasticity of lymphocyte behaviour in response to the ECM.

## Introduction

Lymphocytes continuously patrol in tissues and are recruited to infected areas to detect and clear the area of pathogens and cancer cells. Theoretical studies have shown that the efficiency by which active particles, such as motile cells, can find target particles depends on the characteristics of the trajectories that lymphocytes follow, the local density of the environment, and the distribution of targets. In absence of obstacles, at low target density and if targets need to be revisited multiple times for a 'kill', persistent random walks or random walks characterized by long ballistic phases intermitted by local exploration, such as Lévy walks, perform better than more diffusive, Brownian walks because the ballistic strides prevent local oversampling [1, 2], where the search efficiency depends on the distribution of the stride lengths [3]. For large target densities, Brownian walks become the optimal strategy, whereas for targets that need only a single hit to be killed moving ballistically becomes the optimal strategy [1]. The presence of non-overlapping convex obstacles does not affect the efficiency of Lévy walks, but in porous media characterized by concave obstacle boundaries, such as dense biological tissues, Brownian-like search strategies can become more effective because the ballistic phase of Lévy walks leads to frequent collisions of the particles with the obstacle walls [4]. Relative to Brownian walks, subdiffusive random walks are characterized by enhanced local exploration. This enhances the probability that the active particle binds the target within a given time [5], suggesting that this strategy becomes effective when the lymphocyte has detected its target, e.g., through detection of diffusive signals, but is still unable to bind it, or when the lymphocyte needs to hit the target multiple times for an effective kill of the target [6]. For a recent in-depth study of search efficiencies of subdiffusive, diffusive and superdiffusive random walkers, we refer to Ref. [7].

Consistent with theoretical predictions of the optimal search strategies in presence of obstacles, within densely populated lymph nodes, T lymphocytes perform Brownian walks [8, 9] or persistent random walks [10]. In brain tissue, which is less dense than lymph node tissue, T cells perform Lévy walks [11]. In pancreatic islets, CD4+ T cells perform subdiffusive random walks, whereas CD8+ T cells perform confined random walks [12]. Thus, the characteristics of the trajectories taken by immune cells and the density of the tissues through which they travel determine together how efficiently lymphocytes can find their target cells. Therefore, it is key to understand what cellular properties give rise to the characteristics of the random walks performed by lymphocytes.

Mathematically, the superdiffusive and subdiffusive search strategies observed in leukocytes are non-Markovian and therefore require a memory of previous cell positions (Ref [13] and references therein). Such positional memory can be provided by local modification with the micro-environment, e.g., through autochemotaxis [13], by local modification of the ECM, or by intracellular memory effects such as cell polarization and cell-matrix attachments, as we show here. In agreement with these theoretical considerations, experimental work has shown that plasticity of lymphocyte motility behaviour is dictated both by environmental factors and by cell intrinsic features [14, 15]. An *in vitro* study has shown that the type of extracellular matrix (ECM) used as cell culture substrate affects the motility patterns of B lymphocytes, possibly due to the attachment strength [16]. On fibronectin, B lymphocytes show higher diffusivity and more effective displacement than on collagen IV substrates where cells move more slowly. The B lymphocytes form larger adhesive connections with fibronectin than with collagen IV, and on fibronectin the cells change shape more rapidly than on collagen IV. Similar effects have been found for T lymphocytes. The majority of cells on a casein substrate move through multiple, distinct and temporary adhesion zones, i.e., walking motility, whereas on ICAM-1 substrates, the majority of cells make one continuous contact zone with the substrate, i.e., sliding motility [17]. Apart from these environmental effects, cells also show large individual variation in their motility patterns indicating that cell intrinsic features matter as well. On fibronectin, individual B lymphocytes showed either floating-like behaviour with little attachment, dynamic attachment leading to stepping/walking behaviour, or sustained attachment leading to cells pivoting around their adhesive area [16]. Similarly, T lymphocytes can show either walking, mixed or sliding behaviour, with relative frequencies depending the type of culture substrate [17].

It is still poorly understood what causes, on the one hand, the consistent differences in motility modes between culture substrates, and on the other hand, the large individual differences between cells on the same type of substrate. Modelling helps shed light into such questions [18], and indeed previous theoretical studies have provided useful insights. Copos et al. [19] asked what causes the cellular extension and retraction cycles driving the motility of *Dictyostelium discoideum* cells. They modelled *D. discoideum* forward and backward movement in a force-based model. In agreement with preliminary experimental observations, the model predicted that for low densities of adhesion binding sites in the substrate and low adhesion strength, the cells crawled smoothly along the surface, aka gliding motility. For an increased binding site density or stronger adhesion, the cells switched to stepping motility, that is they moved by extending and reducing their length at a reduced speed relative to gliding motility. For the highest adhesion densities or adhesion strengths, the cells became stationary. Thus, this work showed that both the amount and strength of the attachments can determine the cell motility patterns observed in *D. discoideum*. Aranson and coworkers [20, 21] introduced a phase-field model to study how actin polymerization, dynamical adhesion site formation, and substrate compliance together determine the characteristics of cell trajectories. The model predicted gliding motility at high substrate stiffness, high protrusion strength and high formation

rate of adhesions. At intermediate substrate stiffnesses with sufficiently high protrusion strength and intermediate adhesion formation, the cells displayed a stick-and-slip motility. Thus, phase-field models have provided key insights into how the extracellular matrix affects cell motility. However, the high computational costs of phase-field modelling currently limit the production of the large data volumes required for statistical analyses of cell trajectories. A first step in this direction was taken by Yu et al. [22] who have introduced a computationally efficient, coarse-grained model to study long term cell persistence. The model considered spheroid cells with a fixed pool of focal adhesions. These adhesions were assumed to be widely dispersed within the cells for soft substrates and more narrowly dispersed for rigid substrates. The increased persistence times on rigid substrates led to durotaxis, i.e., preferential movement towards stiffer substrates. In their model, Yu et al. [22] assumed a direct dependence of cell persistence on adhesion distribution.

Altogether, the previous experimental and computational work has shown that the mechanisms driving cell motility and the adherence of the cells to the matrix together are key determinants of cell motility patterns, and hence of the search strategies of lymphocytes. However, to precisely relate mechanistic cell characteristics to cellular search strategies, we must be able to explain how the kinetic interplay between cell motility and cell adhesion determines the bulk characteristics of cell motility, such as velocity distributions and the shape of the mean squared displacement curves. To this end, here we introduce a simplified, yet entirely mechanistic mathematical model of cell motility and the adhesive interaction with the ECM. The model is computationally sufficiently fast for the production of the large numbers of predicted, two-dimensional cell trajectories required for statistical analysis. The model is an extension of the Act model [23], a recent extension of the Cellular Potts model, that provides a fast and phenomenological model of actin dynamics. Depending on the parameter settings, this phenomenological model of actin-polymerization-driven cell motility on its own already displays intermittent (stop-and-go) motility and persistent random walks. The model displays universal coupling between speed and persistence [24], as observed in many cell types. Altogether the Act-model provides a biologically-plausible and computationally-efficient starting point for our purposes. In the remaining part of this paper, we increase the complexity of the model step by step, to show how each component contributes to the final behaviour of the model. First we show that a combination of coarse-grained actin dynamics with simplified dynamics of adhesion formation and detachment reproduces both pivoting behaviour and sliding and stepping behaviour, but fails to reproduce the requirement of cell-substrate adhesion for cell motility. We then introduce a further mechanism into the model in which the cell-ECM adhesions promote cell protrusion by inhibiting retrograde actin flow. The extended model suffices to reproduce the three phases of lymphocyte motility on fibronectin [16]. Thus, our work shows how much of the variety of cell motility can follow from actin dynamics and cell-ECM adhesion, and provides a computationally-tractable model that allows for statistical analysis of cell motility.

## Results

In this section, we first introduce the main assumptions of the model, referring to the Methods section for detail. We then show that this model can reproduce a number of lymphocyte motility modes. Next, we extend the model with feedback from the adhesions onto the actin polymerization force and show that we can capture more dynamic motility behaviours. Overall, our model recapitulates the diversity of lymphocyte motility modes and provides insight into the mechanisms underlying such behavioural diversity.

**Table 1. List of parameters involved in adhesion dynamics and values used for simulations.**

| Parameter | Description | Values | | |
|---|---|---|---|---|
| | | **Figs 2, 3 and 4** | **Figs 6 and 7** | **Fig 8** |
| $\lambda_{Act}$ | Weight of the Act-extension, the maximum protrusive force induced by actin polymerisation | 240 | 120, 240 | 240 |
| $\mathrm{Max}_{Act}$ | Maximum value of the Act-field, actin lifetime | 120 | 120 | 120 |
| (value in Eq 10) | Fraction of $\mathrm{Max}_{Act}$ above which adhesion formation is possible | 0.75 | 0.75 | 0.75 |
| $p_s$ | New adhesion formation rate | 0.004–0.020 | 0.001–0.004 | 0.003,0.001 |
| $p_e$ | Rate of neighbouring grid site to become adhesion site if not already so | 0.0055 | 0.0055 | 0.0015,0.004 |
| $p_d$ | Rate of unbinding adhesion site dependent on adhesionless neighbouring sites | 0.0008 | 0.001 | 0.0004 |
| $\lambda_{adh}$ | Energy required to rupture adhesion upon retraction of the cell | 20–100 | 20–100 | 60 |
| $f$ | Prefactor for the adhesion feedback onto Act model | - | interval $[b,1]$ | interval $[b,1]$ |
| $b$ | Base value of $f$ in absence of adhesions | - | 0.5 | 0.5 |
| $s$ | Adhesion area fraction saturation threshold above which $f = 1$ | - | 0.1 | 0.12 |

## Modelling cell-matrix adhesions

The computational model is based on the Act model [23, 24], which is an extension of the Cellular Potts model (CPM, [25, 26]) that efficiently simulates persistent, amoeboid cell motility emerging from feedback mechanism inspired by detailed insights into actin-polymerisation driven cell motility. Depending on its parameter settings, the model produces a variety of realistic cell shapes, and reproduces realistic cell polarisation and cell trajectories. In short, the model is two-dimensional, and can be interpreted as a projection of the three-dimensional cell and the underlying substrate from the top. The Act extension keeps track of recent "actin polymerisation" inside the cell, which is represented through *Act values* at each lattice site. Novel protrusions get a high *Act value* and cell protrusions at sites with locally high *Act values* are favoured. Two important parameters for this are $\lambda_{Act}$, the weight of the Act model that can be interpreted as the maximum protrusive force of the actin network, and $\mathrm{Max}_{Act}$, the maximum Act value, interpretable as the lifetime of an actin subunit within the actin network (see Table 1). Here we extend the model with dynamical cell-matrix adhesions (Fig 1). Full detail is given in the Methods section.

In this two-dimensional model we consider integrin bonds between the cell and the underlying substrate, by which the cell adheres to the substrate. They are represented by a binary number in each lattice site, which indicates if at this site there is an active adhesion or not. In biological cells, the formation of new adhesions occurs at the cell's leading edge during actin polymerisation and pseudopod protrusion [27–29], as well as through formation and expansion of adhesions in the middle region of the cell. In absence of forces pulling the whole cell off of the substrate, due to the two-dimensional representation of the cell, only the adhesions at the edges of the cell will affect cell motility. Nevertheless, we will need to consider formation and degradation of adhesion in the middle of the cell as well, because–as will become apparent when we consider the dynamics of the model–a large patch of adhesions at the cell edge will still resist a series of unbinding events better than a few isolated adhesions. In addition, in an extended version of the model (see Section Including the effects of cell-substrate adhesion on cell protrusion) we consider a feedback of the adhesions on actin polymerization.

We mimic formation of adhesions at the leading edge as follows: in the Act model, the leading edge is marked by lattice sites with high Act values. We thus assume that the probability of adhesion formation is proportional to the Act-value. More precisely, defining $V(\vec{x}) = \{\vec{y} \in \mathrm{NB}(\vec{x})|\sigma(\vec{y}) = \sigma(\vec{x})\}$ as the Moore neighbourhood of lattice site $\vec{x}$ ($\mathrm{NB}(\vec{x})$) restricted to the same cell as site $\vec{x}$, a lattice site $\vec{x}$ receives an adhesion with probability $p_s$ (Fig 1A and 1B,

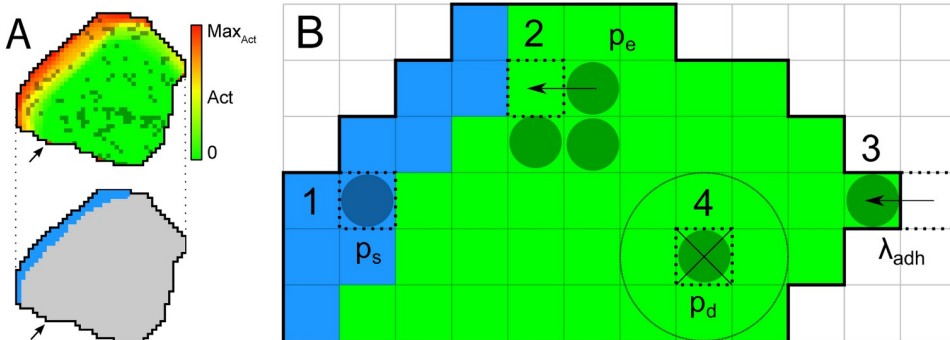

**Fig 1. Overview of the adhesion processes within the model.** A) Top: overview of a simulated cell. Red to yellow shading indicates the Act-level of each lattice site according to the colour bar. Darker coloured lattice sites contain an adhesion. Bottom: the same cell with in blue the region where new adhesions can form, as the local Act-levels exceeds the $0.75 \, \text{Max}_{Act}$ threshold. Both: Arrows point to area with one lattice site with high Act-level due to a recent extension of the cell (top, red), but the geometric mean of Act-levels does not exceed the threshold and hence new adhesions cannot form there (bottom, grey). B) Visual summary of adhesion processes. Dark coloured circles indicate lattice sites containing an adhesion. 1) New adhesions can form spontaneously with probability $p_s$ at cell lattice sites where the geometric means of Act exceeds the threshold of $0.75 \, \text{Max}_{Act}$ (blue region). 2) An adhesion patch can grow by Eden growth. A random neighbour of an adhesion site is selected. When it does not contain an adhesion yet, the patch extends into that lattice site with probability $p_e$. 3) To break an adhesion, retractions must overcome an energy cost $\Delta H = \lambda_{adh}$. 4) Adhesions can also unbind spontaneously, depending on the number of neighbouring lattice sites without adhesions and probability $p_d$.

process 1) if the geometric mean of Act-value $Act(\vec{y})$, with $\vec{y} \in V(\vec{x})$ exceeds a threshold $\left(\prod_{\vec{y} \in V(\vec{x})} Act(\vec{y})\right)^{\frac{1}{|V(\vec{x})|}} \geq 0.75 \, \text{Max}_{Act}$. Note that in our model, $p_s$ lumps together the effect of multiple biological processes, including the rate at which integrin molecules bind to their ligands in the extracellular matrix, and the concentration of integrins at the cell front. A detailed biophysical model suggests that local integrin-substrate bonds favour the formation of adjacent bonds, because they stabilise the membrane, reducing membrane fluctuations [30]. We simplify patch expansion phenomenologically using the Eden model [31] of radial colony growth. Empty lattice sites adjacent to an existing adhesion site join the adhesion patch with probability $p_e$ (Fig 1B, process 2). Unbinding of cell-matrix adhesions occurs spontaneously or as a result of cellular forces. More specifically, adhesion patches rupture from the edges of the patches [17]. We model this process only at the edge of the cell, where cellular contraction forces are sufficient to break bonds. Integrin bonds are known to show catch-slip bond behaviour, meaning that initially the bond strengthens with increase of force, but will still break if enough force is applied. Here, we neglect this specific behaviour and associate a single required energy cost of $\lambda_{adh}$ with the rupture of adhesions at the retracting edge (Fig 1B, process 3). $\lambda_{adh}$ is given by the binding affinity between integrins and their ligands and the concentration of integrins bound to the ECM. In addition, we assume that association with adjacent adhesions reduces the spontaneous unbinding rate. The probability that an adhesion site unbinds thus becomes $p_d \cdot \left(\frac{|\{\text{nb} \in \text{NB}(\vec{x}) \setminus \{\vec{x}\} | Adh(\text{nb}) = 0\}|}{|\{\text{nb} \in \text{NB}(\vec{x})\} \setminus \{\vec{x}\}|}\right)^2$, where $\text{NB}(\vec{x})$ indicates the Moore neighbourhood of lattice site $\vec{x}$ (Fig 1B, process 4). Our model thus represents the dynamic behaviour of cell-matrix adhesions.

All in all, our proposed model extension for adhesions is relatively simple and computationally light. All adhesion dynamics are governed by the four parameters $p_s$, $p_e$, $p_d$ and $\lambda_{adh}$. An overview of all the relevant parameters is shown in Table 1. Estimates of length scale, time scale, and the parameter $\text{Max}_{Act}$, as well as the relative magnitudes of $\lambda_{Act}$ and $\lambda_{adh}$ can be found in S1 Text. The simulations were performed using Tissue Simulation Toolkit ([32];

source code in S1 Data). An interactive version in Artistoo [33] of the model running in a web browser is in S2 Data.

## Model reproduces walking motility and sustained attachment leading to pivoting motility

We first investigated the influence of the cell adhesion formation probability $p_s$, and adhesion strength $\lambda_{adh}$ on the motility patterns. Fig 2A–2D summarize the variety of patterns observed in the model. We observe roughly two types of cell motility patterns. If $\lambda_{adh}$ is low to moderate (Fig 2A and 2C, S1 Video) the substrate adhesions do not affect retractions of the cell, and the model behaves, therefore, practically like the standard Act-model [23] which mostly reproduces sliding motility [24] For larger values of $\lambda_{adh}$, the simulated cells show less persistent motility (Fig 2B and 2D, S1 Video). For high values of $\lambda_{adh}$ and $p_s$ cells form sustained adhesions. For these parameter values, the adhesions form easily and require much energy to break,

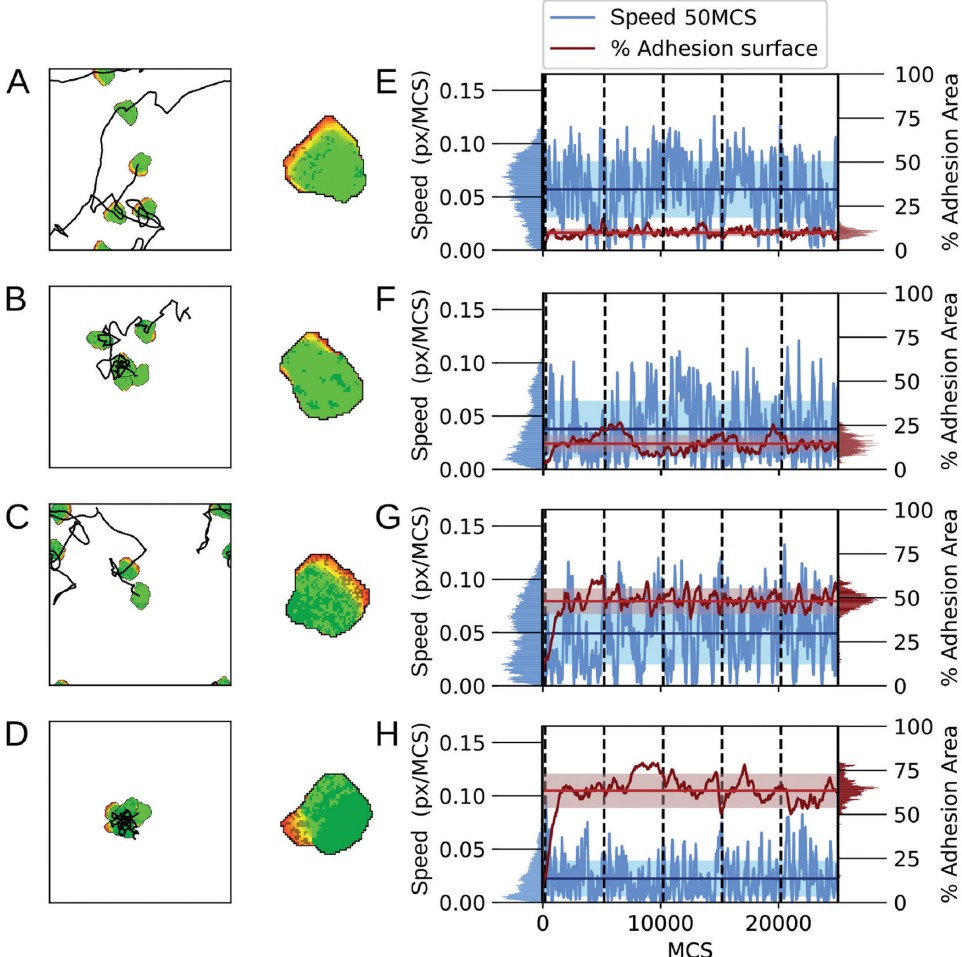

**Fig 2. Simulations of the model showing different motility types.** A-D) First column, a display of a single cell at 5000 MCS interval snapshots combined with the cell centre's trajectory. Each trajectory starts in the centre of the field and periodic boundaries are used. Second column, a close-up of the cell with the adhesions displayed in a darker colour. E-H) A plot of the cell's speed and percentage of the cell's area containing adhesions corresponding to the track on the left. Vertical dashed lines indicate the times of the snapshots on the left. Parameters are: A,E) $\lambda_{adh} = 20$, $p_s = 0.004$, B,F) $\lambda_{adh} = 100$, $p_s = 0.004$, C,G) $\lambda_{adh} = 20$, $p_s = 0.02$, D,H) $\lambda_{adh} = 100$, $p_s = 0.02$. Furthermore, $p_d = 0.0008$ for all cases. These simulations are also available as S1 Video.

such that cells will remain stuck in place on the matrix. However, they can still make protrusions around them, leading to pivoting motility (Fig 2D, S1 Video).

To further characterize the cell motility and the effect of the dynamic adhesion patches, we measured the cell speed alongside with the summed area of the adhesions divided by the total area of the cell (Fig 2E–2H). Note that this percentage adhesion area should be interpreted as the percentage of the bottom area of the cell that is adhered to the underlying substrate. First we looked at the effect of $p_s$ and $\lambda_{adh}$ independently, expecting that $p_s$ regulates the adhesion area. Indeed, at relatively low values of $p_s = 0.004$ (Fig 2E and 2F), the cells formed low adhesion area, and at higher values of $p_s = 0.02$, we observed a larger adhesion area (Fig 2G and 2H). $p_s$ thus positively correlated with the percentage of adhesion area. We expect that $\lambda_{adh}$ regulates cell speed. Indeed, at relatively low values of $\lambda_{adh} = 20$ (Fig 2E and 2G) the average speed was higher and also fluctuated less than at higher values of $\lambda_{adh} = 100$ (Fig 2F and 2H), as indicated by the mean squared logarithmic difference (E:0.00313 and G:0.00442 versus F:0.0072 and H:0.0167; see distributions at the left). $\lambda_{adh}$ thus negatively correlated with cell speed. Interestingly, in Fig 2B and 2F we observed sliding-stepping-like behaviour: the cells frequently slowed down followed by burst of acceleration. However, the simulations did not display the cyclic expansive and contractile shape changes observed in experiments [16] and reproduced in more complex models [19].

Overall, the behaviour of our model resembles observations by Rey-Barroso et al. [16] that B cells on fibronectin with fluctuating adhesion areas showed walking behaviour, and cells with large and sustained adhesion surfaces displaced very little and the adhesion patch did not displace. The behaviour of the model also agrees with observations of Jacobelli et al. [17] that T cells displaying a gliding motility with higher adhesion area have lower speed than cells with a walking motility with lower adhesion area. An important difference is that the gliding cells in experiments show a large adhesion patch at the front where they assist in cell protrusion, whereas in our model the bulk of the adhesions appears more to the back, where they mostly reduce cell retraction probability (Fig 2C and 2G).

## Adhesions slow down cell motility and reduce diffusivity

The examples shown in Fig 2 and S1 Video suggest that higher adhesion area correlates with reduced cell speed and reduced cell diffusivity. To characterize this potential correlation, we analysed 1000 independent runs for different combinations of $p_s$ and $\lambda_{adh}$, and measured the average cell speed and adhesion area. Fig 3A shows the mean speed of the cells as a function of $p_s$ and $\lambda_{adh}$, and Fig 3B shows the diffusivity. In both panels, symbols with the same value of $\lambda_{adh}$ are connected and symbols of the same colour have the same value of $p_s$. Fig 3A shows that increasing the value of $\lambda_{adh}$ decreases cell speed and that the average adhesion area increases with the value of $p_s$. Furthermore, for high values of $\lambda_{adh}$, the effects of $p_s$ on cell adhesion are larger, and for high values of $p_s$, the effects of $\lambda_{adh}$ on cell speed are also larger. Fig 3B plots the diffusivity of the cells and adhesion area as a function of $p_s$ and $\lambda_{adh}$. As a crude measure for diffusivity we took the slope of the mean squared displacement curves from 11500 to 24000 MCS, assuming the MSD has entered a linear regime in that time period as discussed in more detail below. Similar to the cell speed, increasing $\lambda_{adh}$ results in lower diffusivity and the effect is larger for higher values of $p_s$. Moreover, the effect of $\lambda_{adh}$ is larger on cell diffusivity than on cell speed. The drop in instantaneous cell speed (from highest to lowest in Fig 3A a reduction to about one half) is modest compared to the drop in diffusion coefficient (Fig 3B, from highest to lowest a reduction to about one fiftieth). Cell diffusivity is determined by instantaneous cell speed and persistence of movement direction. Thus, this observation suggests that the drop in diffusivity is for a large part caused by a loss of cell persistence.

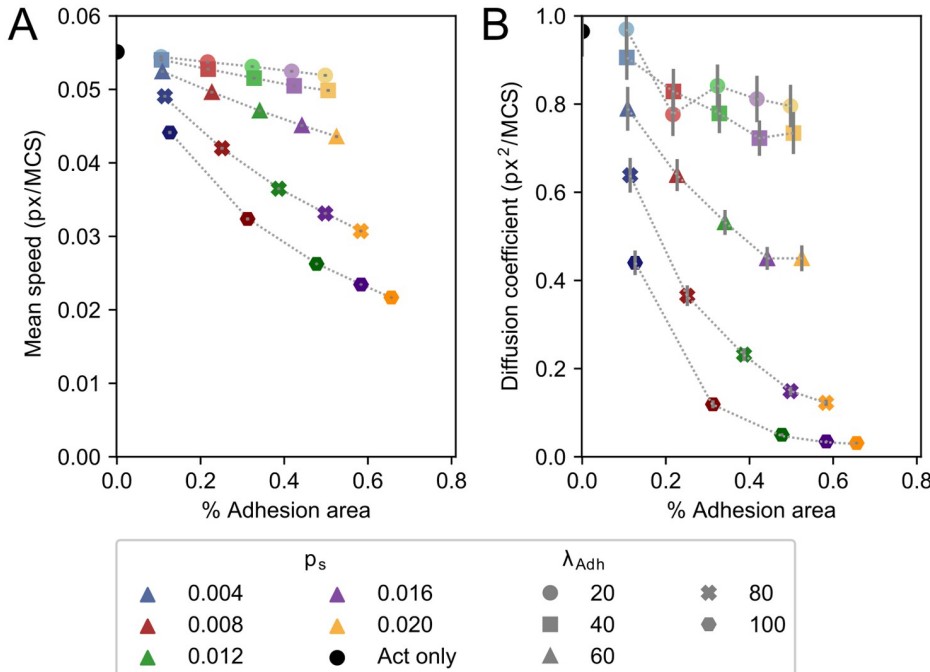

**Fig 3. Mean speed, diffusion coefficient, and mean adhesive area change by increasing $p_s$ and $\lambda_{adh}$.** Mean speed (A) and diffusion coefficient (B) plotted against mean percentual adhesion area for different values of parameters $p_s$ and $\lambda_{adh}$. Each dot represents the mean of 1000 independent simulations. Different colours indicate different $p_s$, where shades from light to dark and marker symbol indicate $\lambda_{adh} \in \{20, 40, 60, 80, 100\}$. For reference, the mean speed and diffusion coefficient of the Act model without any adhesions are indicated by a black circle. Error bars indicate 95% confidence intervals.

## Reduced diffusivity of adhesive cells is due to reduced persistence times

To further characterize the cause of the reduced diffusivity at high values of $p_s$ and $\lambda_{adh}$, we fitted the mean squared displacement curves with a persistent random walker model, the Fürth equation [34, 35]:

$$MSD(t) = 4\frac{v_{th}^2}{\gamma_1^2}(\gamma_1 t - 1 + e^{-t\gamma_1}),\tag{1}$$

where $v_{th}$ is the walker's speed and $\gamma_1$ is its persistence time (S1 Fig). The persistent random walker model fits well with the MSD curves at long time scale, but it fails at the short time scale where the cell trajectories are mainly determined by the random fluctuations in the CPM. We therefore extended Eq 1 with a term for translational diffusion [36],

$$MSD(t) = 4\frac{v_{th}^2}{\gamma_1^2}(\gamma_1 t - 1 + e^{-t\gamma_1}) + D_T t\tag{2}$$

The extended Fürth equation (Eq 2, S1 Fig) gives good fits for most cases (Fig 4A–4C). Extending the Fürth equation with anomalous diffusion instead of translational diffusion, i.e., adding the term $D_T t^\beta$, does not result in better fits, despite the additional parameter (S1 Fig). The persistence times, as obtained from the extended Fürth equation (Eq 2), indicate that indeed the reduced diffusivity can be attributed for a large part to a reduced persistence time with higher adhesion energies and large adhesive areas (S2 Fig). However, for the highest values of $\lambda_{adh} = 80$ to $\lambda_{adh} = 100$ Eq 2 still does not fit well with the data (Fig 4D). We attempted

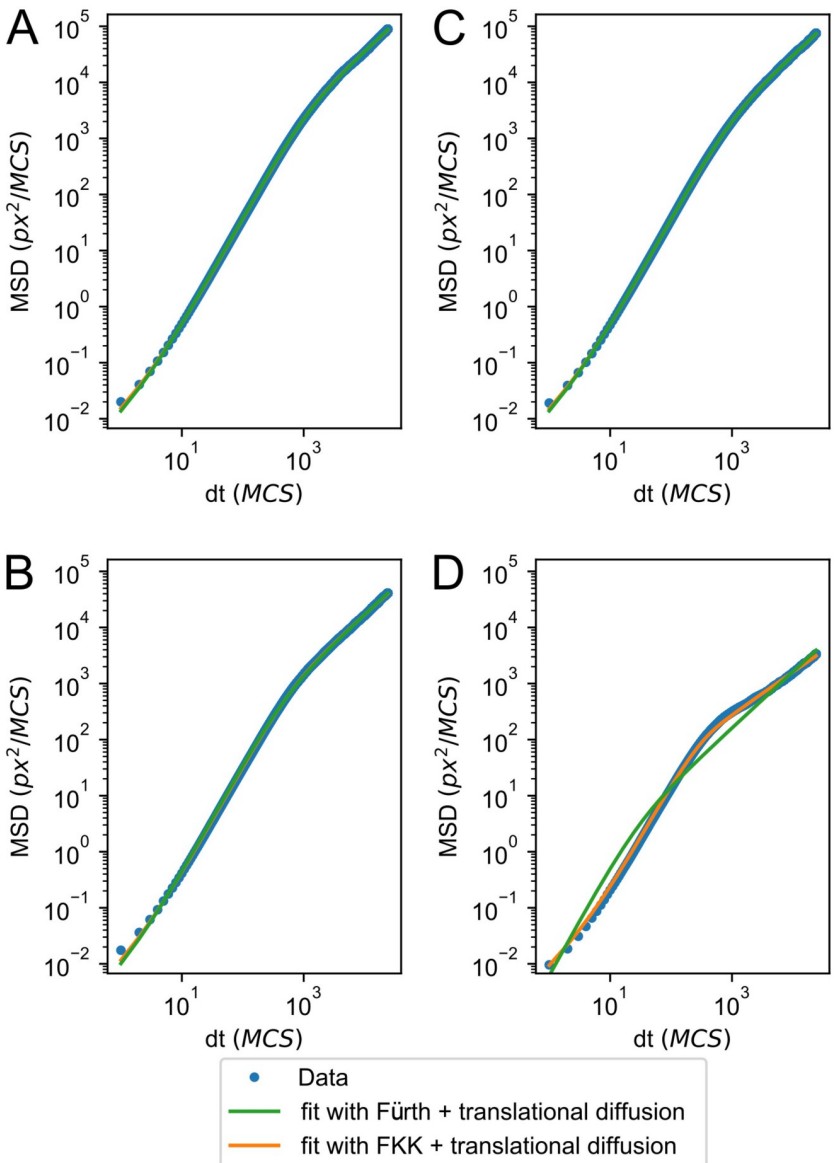

**Fig 4. MSD fits either a persistent random walker or a subdiffusive persistent random walker.** Log-log plot of MSD for the four scenarios in Fig 2, with fits of Eqs 2 and 4. Parameters are: A) $\lambda_{adh} = 20$, $p_s = 0.004$, B) $\lambda_{adh} = 100$, $p_s = 0.004$, C) $\lambda_{adh} = 20$, $p_s = 0.02$, D) $\lambda_{adh} = 100$, $p_s = 0.02$.

to improve the fit by increasing the initialization period left out of the MSD computation, in order to compute the MSD of cells closer to their dynamic equilibrium in both Act model dynamics as well as adhesion-extension dynamics. This barely improved the fit and suggests that cell motility in this regime cannot be correctly described by a persistent random walker with translational diffusion.

## Strongly adhesive cells show subdiffusive behaviour

To characterise the diffusive behaviour of strongly adherent cells at high $\lambda_{adh}$ values we fitted the data with the MSD of a fractional Klein-Kramers (FKK) process. Previously the fractional

Klein-Kramers (FKK) process was proposed to describe the motility of transformed Madin-Darby canine kidney cells [37]. The MSD of the FKK process was given by,

$$MSD(t) = 4v_{th}^2 t^2 E_{\alpha,3}(-\gamma_\alpha t^\alpha) + (2\eta)^2, \tag{3}$$

where $E_{\alpha,3}$ is the generalized Mittag-Leffler function and $\eta$ is a noise term. The standard Fürth model (Eq 1) is a special case of the FKK-process for $\alpha = 1$ and $\eta = 0$, where $\alpha$ parameterizes the long-term diffusive behaviour and is restricted to $0 < \alpha \leq 1$ [38]. The same MSD expression, except for the noise term, has also been derived from a fractional Langevin equation of motion, which only holds for $1 < \alpha < 2$ [39]. Since we already concluded that translational diffusion plays a significant role in the short-time scale of the CPM, we assume that the noise term is due to translational diffusion, thus obtaining,

$$MSD(t) = 4v_{th}^2 t^2 E_{\alpha,3}(-\gamma_\alpha t^\alpha) + D_T t. \tag{4}$$

which reduces to Eq 2 for $\alpha = 1$. For $t \to \infty$, Eqs 3 and 4 can be approximated by $MSD(t) \sim \frac{4D_{th}t^{2-\alpha}}{\Gamma(3-\alpha)}$ [37]. So for $\alpha > 1$, the long-term behaviour is subdiffusive, whereas for $\alpha < 1$, the long-term behaviour is superdiffusive.

In the cases where Eq 2 fits well, we obtain $\alpha \approx 1$ (Table 2). However, for the cases were Eq 2 fits badly, Eq 4 has a better fit and $\alpha > 1$ (Fig 4, Table 2) indicating subdiffusive behaviour. This behaviour corresponds to cases where the cells are strongly attached to the matrix, pivoting around their adhesion patch. Thus, these cells move persistently on a local scale as they have a single protrusion front, but they move subdiffusively on a longer timescale as they stay within a confined area.

## Including the effects of cell-substrate adhesion on cell protrusion

So far, the model can explain (i) sliding and stepping, and (ii) pivoting cell behaviour as observed in Ref. [16]. However, key differences between the behaviour of the extended Act-model with real cell motility remain. In particular, in the extended model the cell-substrate adhesions only reduce the probability of cell retraction, whereas in biological cells, the adhesions near the cell front are instrumental for pulling the cell forward. For this reason, the current model still fails to explain (iii) the floating phase that was also observed in Ref. [16], i.e., the observation that B cells with a low adhesive area or no adhesive area on a fibronectin substrate show low displacement compared to cells with dynamic attachment [16].

To also recapitulate such floating behaviour, we extended the model as follows. In presence of a stable adhesion, the forces generated by actin polymerization are transferred onto the matrix leading to protrusion. With reduced adhesions, actin polymerisation more often leads to treadmilling. Thus, in presence of cell-matrix adhesion, actin polymerisation more efficiently translates to cell protrusion [40–42]. We model this effect using a prefactor $f$ to $\lambda_{Act}$, such that it dynamically alters the strength by which the Act-values affect the cell protrusions, and hence the protrusion force. For simplicity, the protrusion efficiency increases linearly with

**Table 2. Fitted values of $\alpha$ from Eq 4 for different values of $\lambda_{adh}$ and $p_s$.**

| Parameters | $\alpha$ |
|---|---|
| $\lambda_{adh} = 20$, $p_s = 0.004$ | 1.019 |
| $\lambda_{adh} = 20$, $p_s = 0.020$ | 1.013 |
| $\lambda_{adh} = 100$, $p_s = 0.004$ | 1.024 |
| $\lambda_{adh} = 100$, $p_s = 0.020$ | 1.257 |

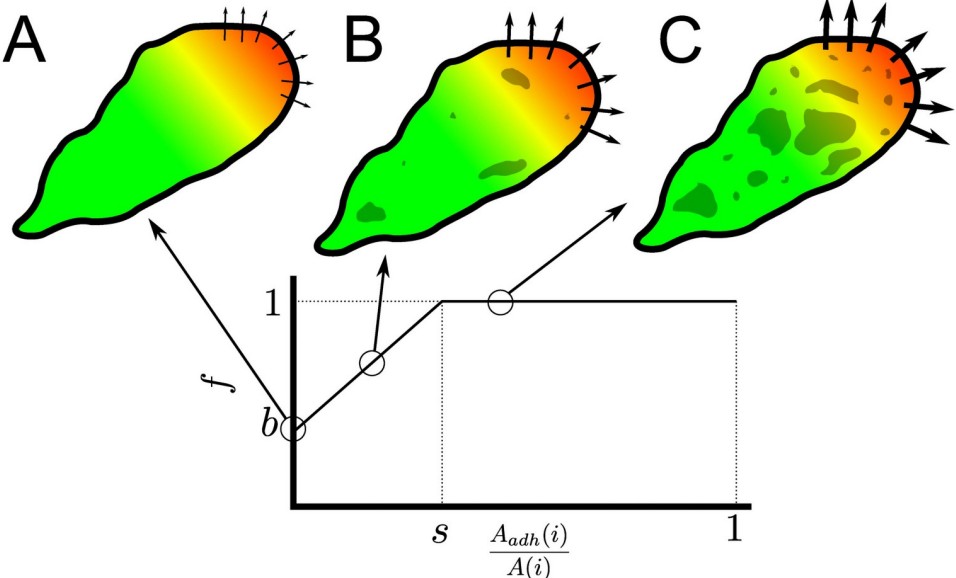

**Fig 5. Schematic representation of the effect of adhesion on cell propulsion strength.** Colour schemes are similar to Fig 1A. Arrow width corresponds to the effective protrusion strength $f\lambda_{Act}$. A) In the absence of adhesions, the propulsion prefactor $f$ is equal to the base level $b$. B) Below the saturation point $s$, $f$ increases linearly with adhesion area. C) Above the adhesion area saturation point $s$, prefactor $f$, and thus effective protrusion strength $f\lambda_{Act}$, are maximal.

the cell's total adhesion area. Furthermore, we assume that there is a threshold adhesion area at which the adhesive force suffices to withstand the forces of actin polymerization. Hence, we define,

$$
f = \begin{cases} b + \dfrac{1-b}{s}\dfrac{A_{adh}(i)}{A(i)} & \text{if } \dfrac{A_{adh}(i)}{A(i)} \leq s \\[2ex] 1 & \text{if } \dfrac{A_{adh}(i)}{A(i)} > s \end{cases} \tag{5}
$$

with $b$ the baseline protrusion efficiency and $s$ the threshold adhesion area. Thus the effect of the adhesion area on the propulsion strength only affects cell motility if the adhesion area is below or near the threshold $s$. A schematic overview is shown in Fig 5.

### Extended model reproduces all three phases of cell motility

To test if the new model indeed reproduces floating motility alongside the other two phases of motility observed in Ref. [16], we focus on parameter combinations that result in adhesion areas below or around the threshold $s$. For adhesion areas above the threshold $s$, the model behaviour does not change. We choose $s = 0.1$ and a baseline protrusion efficiency $b = 0.5$. From the previous section, we know that $p_s$ is the main parameter controlling adhesion area, so we chose $p_s \leq 0.004$.

The model shows a variety of behaviours depending on the value of $p_s$ (Fig 6, S2 Video). For very low values of $p_s = 0.001$ (Fig 6A), cells make only a small number of tiny adhesion patches and thus have a small adhesive area. Furthermore, they explore relatively small areas (Fig 6A and 6B). Nevertheless, their trajectories can still be described well with Eq 2 or with Eq 4 with

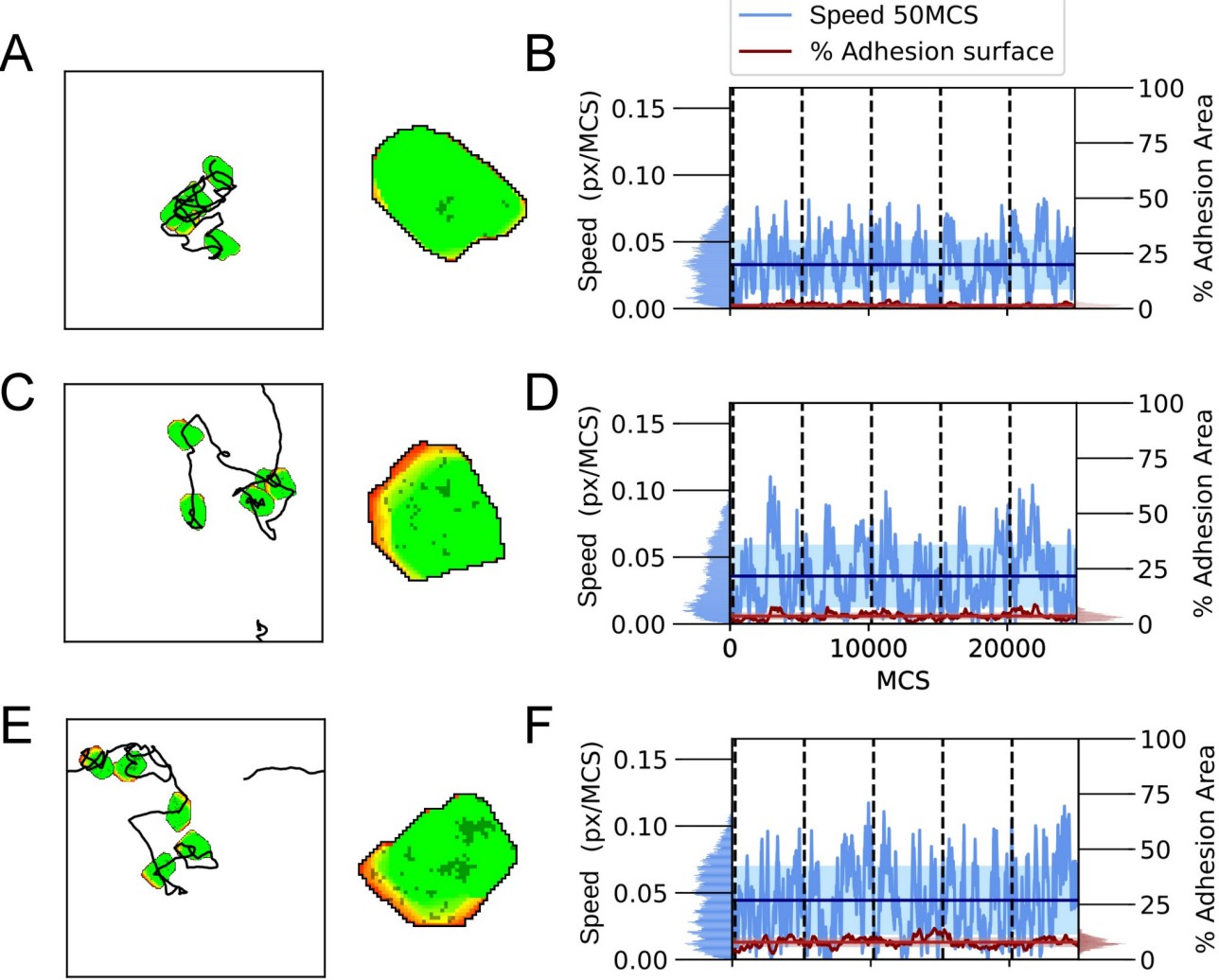

**Fig 6. Simulations of the model with adhesion-propulsion feedback.** A-C) First column, a display of a single cell at 5000 MCS intervals combined with trajectory of the cell centre. Each trajectory starts in the middle and periodic boundaries are used. Second column, a close-up of the cell with the adhesion displayed in a darker colour. D-F) A plot of the cell's speed and percentage of the cell's area containing adhesions corresponding to the track on the left. Parameters are: A,D) $\lambda_{adh} = 100$, $p_s = 0.001$; C,F) $\lambda_{adh} = 100$, $p_s = 0.0025$; B,E) $\lambda_{adh} = 100$, $p_s = 0.004$. Furthermore $p_d = 0.008$ for all cases. These simulations are also available as S2 Video.

$\alpha = 0.974$, so the type of motility can still be classified as a persistent random walk, albeit with lower persistence time (S3 Fig). For increased adhesion formation probability $p_s = 0.0025$, stick-slip behaviour is observed (Fig 6C). Interestingly, in contrast to the initial model, the cells accelerate as the adhesion areas grow and they decelerate when they have lost their adhesions (Fig 6D). For $p_s = 0.004$ (Fig 6E), the mean adhesive area approaches the threshold. In this case, the diffusivity and persistence are lower compared to the model without the adhesion-protrusion feedback, but the cell speed is comparable (Figs 6F and 7A).

Fig 7 shows the average speed (Fig 7A) and diffusivity of cells (Fig 7B) with low adhesive areas for both the initial (filled symbols) and the extended model (open symbols). The cell speed and cell diffusivity of the standard Act model are shown in the figure for reference. The cell speed and cell diffusivity of the initial model follow the same trend as observed in Fig 3,

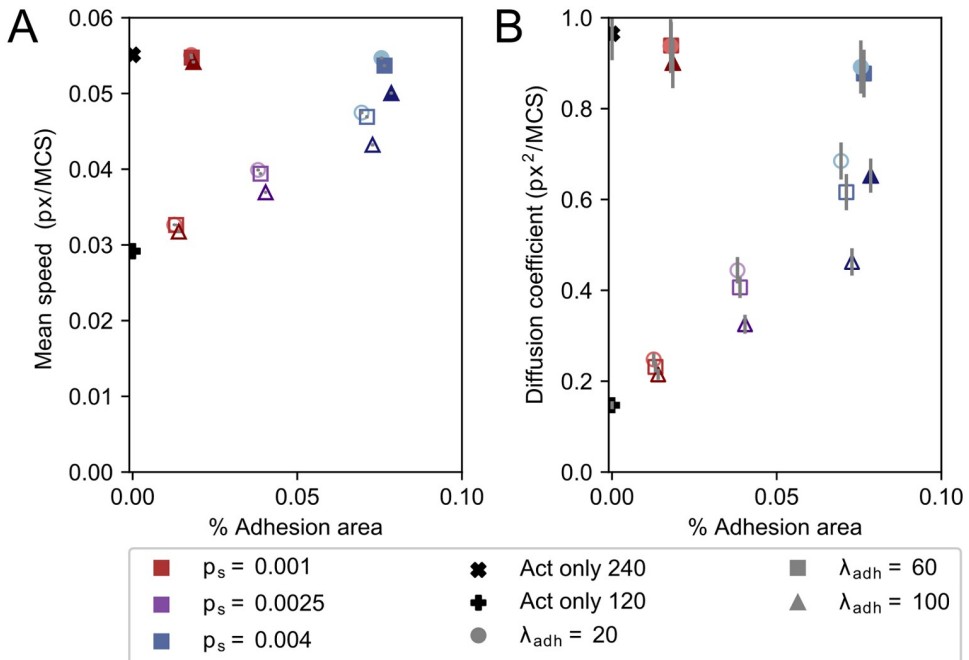

**Fig 7. Mean speed, diffusion coefficient and adhesion area differ between the model with and without adhesion-propulsion feedback.** Mean speed (A) and diffusion coefficient (B) plotted against mean percentual adhesion area for different values of parameters $p_s$ and $\lambda_{adh}$. Each dot represents the mean over 1000 independent simulations. Filled symbols are results from the model without adhesion-propulsion feedback, and open symbols show results from the model with adhesion-propulsion feedback. Different colours indicate different $p_s$, where shades from light to dark and marker symbol indicate $\lambda_{adh} \in \{20, 60, 100\}$. Error bars indicate 95% confidence intervals.

both decreasing for increasing values of $p_s$ and $\lambda_{adh}$. The model converges to the Act model for low adhesion areas as expected ($\lambda_{Act} = 240$ (black cross)). Similarly, the extended model converges to the Act model for low adhesion area (black plus sign), where $\lambda_{Act} = 120$ corresponds to the baseline propulsion strength $b = 1/2$. Near the adhesion area threshold of $s = 0.1$ both models show the same behaviour. The mean speed and diffusion coefficient increase monotonically between these two extremes: higher adhesion areas lead to higher speed. Whereas the value of $\lambda_{adh}$ has little effect on the cell speed in this parameter regime, $p_s$ determines the adhesion area and thus has a large effect on cell speed. Remarkably, there is a slight difference in mean adhesion area between the models. This small effect is likely due to the positive feedback loop between adhesion growth and cell propulsion in the extended model.

In conclusion, by adjusting the parameters $p_s$ and $\lambda_{adh}$, the extended model provides a minimal model explaining the three motility phases of B cells observed on fibronectin: for low cell-matrix attachment the cells have reduced motility (floating motility), for increased cell-matrix attachment the cells form dynamic adhesion areas and display increased motility (sliding-stepping motility), and for the strongest cell-matrix attachment the cells displayed sustained attachment with low displacement (pivoting motility) [16].

## Distribution pattern of adhesions influences cell motility by changing the persistence times

So far, we have neglected the effect of adhesion distribution and variation in time on cell motility. However, the dynamics of the distribution and size of the adhesive patches over the cell

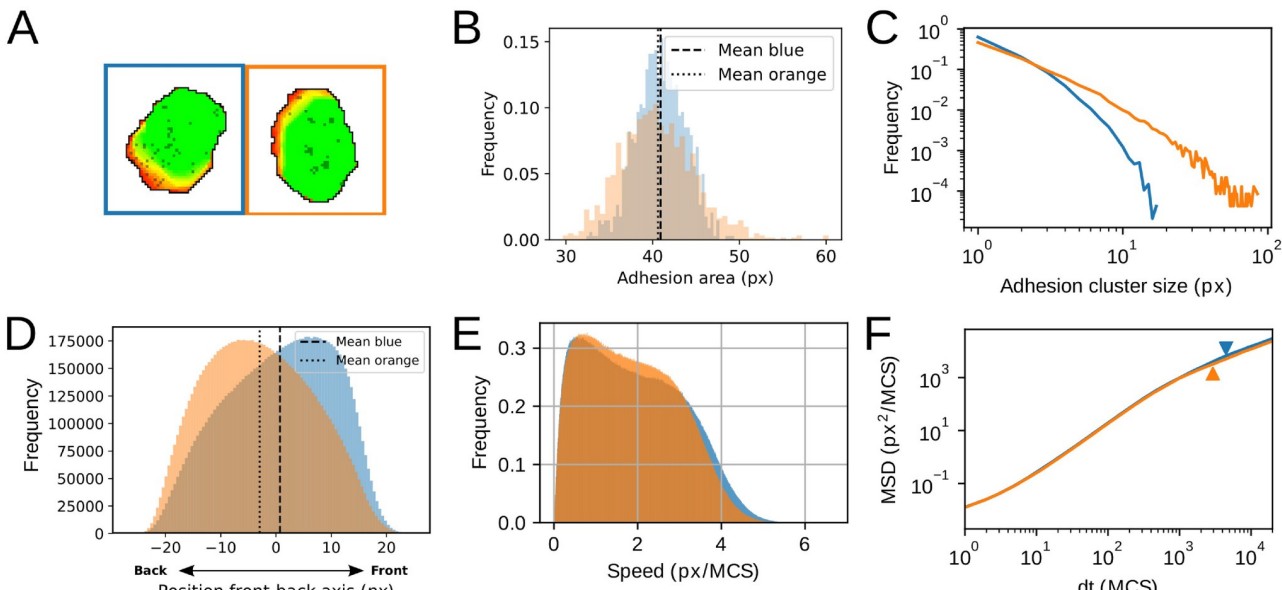

**Fig 8. Adhesion growth dynamics influence adhesion cluster size and localisation, cell speed and MSD.** (A) Example of different adhesion cluster size for different parameter values of $p_s$ and $p_e$. Blue: $p_s = 0.003$, $p_e = 0.0015$ resulting in multiple small adhesions. Orange: $p_s = 0.001$, $p_e = 0.004$, resulting in a small number of larger clusters. Colours in B-F correspond to these parameter settings. (B) Average adhesion area of 1000 cells over time measured after a 'burn in' time of 1000 MCS; each data point represents one cell and vertical lines show the average. (C) Distribution of cluster sizes for 1000 independent simulations for each parameter setting on a logarithmic scale. Distribution of blue does not exceed cluster size 20. (D) Distribution of adhesions along the front-rear axis of the cell for 1000 independent simulations for each parameter setting. Axis was determined from the cell's position at 25 MCS in the past and 25 MCS in the future at 100 MCS intervals. Position 0 indicates the cell's centre of mass. Dotted lines show the means of the distributions. (E) Distribution of instantaneous speed of 1000 independent simulations for each parameter setting. Mean speed for orange is lower compared to blue. (F) Log-log plot of the MSD. The onset of the second linear regime (log-log slope approximately equal to 1) is marked with an arrowhead in corresponding colours. This regime starts at smaller *dt* for the orange curve compared to the blue curve, which corresponds to a lowered persistence time compared to blue. Simulations are also available as S3 Video.

may have a large effect on cell motility patterns. For example, the distribution of the adhesion clusters differs between walking and sliding T cells [17]. Sliding T cells have a large contact area at the cell front. Walking T cells, in contrast, had multiple distinct adhesion areas that were distributed over the cells, including the rear end of the cell.

To gain insight in the impact of adhesion patch distribution on cell motility, we explored parameter settings resulting in the same adhesion area but with different adhesion cluster size distributions We did this by varying the formation rate for new adhesions ($p_s$) as well as the adhesion growth rate ($p_e$). Fig 8 shows the results of two such parameter settings resulting in the same time-averaged adhesion area. The blue parameter set obtains its adhesive area mostly through the formation of new adhesions ($p_s > p_e$, Fig 8A: blue), whereas the orange parameter expansion of existing adhesion clusters dominates ($p_s < p_e$, Fig 8A: orange, see also S3 Video). The two parameter sets form cells with approximately the same average adhesion area (Fig 8B; t-Test for difference: p = 0.07), but the cluster size distribution differs (Fig 8C). For $p_s > p_e$ there are many small adhesion clusters (blue line), whereas for $p_s < p_e$ (orange line) small clusters are combined with a few large clusters. Furthermore, the distribution of adhesion along the cell differs (two-sample K–S test $D = 0.164$, $p = 0.0$): if the cell forms many small adhesions (blue), adhesions are located preferentially towards the front of the cell, whereas adhesions are located more towards the back of the cell if the cell forms larger adhesions (orange; Fig 8D).

Next, we studied whether the differences in adhesion distribution lead to differences in cell motility. First, the speed distribution has a slightly higher mean for the many-small-cluster

(blue) setting, and appears more bimodal than the few-large-cluster (orange) setting (Fig 8E). The MSD curves largely overlap on the short term, but on the long term the few-large-cluster (orange) setting shows an earlier start of the final linear regime. The onset of this regime corresponds to the persistence time, which we obtained by fitting the MSD curves with Eq 2 as well as Eq 4. Indeed, the few-large-cluster (orange) setting has about 25% lower persistence time than the many-small-cluster (blue) setting, possibly because it is harder to detach a large adhesion patch than a few small ones. Although we cannot fully exclude that these differences are due to the wider distribution of cluster sizes (Fig 8B), these data suggest that not only the total adhesion area, but also the location and distribution of the adhesion clusters influences cell motility. This further shows that the dynamics of cell-matrix adhesion can be a key component of the plasticity of cell motility.

## Discussion

The variability in cell trajectories of lymphocytes, as observed in both *in vitro* and *in vivo* studies, can be ascribed to both intrinsic and extrinsic factors. More specifically, the dynamics of the cytoskeleton are influenced by internal cellular processes and interactions with the local ECM, thereby shaping cell motility. In order to investigate how extrinsic factors determine cell motion, we present here a CPM model that combines actin dynamics and cell-ECM interactions by expanding the existing Act model with cell-matrix adhesions. In this model, cell-matrix adhesions can form *de novo* if the local actin cytoskeleton is sufficiently established. Subsequent adhesion dynamics are: i) expansion into adhesion patches, ii) shrinkage of adhesion patches, or iii) adhesion breakage at a set energy cost. By combining the actin-dependent cell propulsion of the Act model with cell-matrix adhesions, our model captures key features of cell motility as seen *in vitro*. For example, it can describe the floating, stepping, and pivoting behaviour observed in B cells on fibronectin [16] as well as the walking and gliding behaviour observed in T cells on ICAM and casein [17]. The first by altering adhesion bond energy cost and *de novo* adhesion formation and the latter by adjusting *de novo* adhesion formation and adhesion patch growth.

   In addition to recapitulating hallmark modes of cell motility, our model sheds light on how cell-ECM interactions determine cell motion on longer time scales. For example, increasing the strength of adhesion bonds, $\lambda_{adh}$, and increasing the probability by which adhesions form *de novo*, $p_s$, decreases cell speed and especially cell persistence. Strongly adherent cells primarily pivoted around their axis and, interestingly, such motility was persistent on a short time scale, but subdiffusive on long time scales. To sufficiently capture the behaviour of weakly adherent cells, cell propulsion strength was made dependent on the degree of adhesion in a second extension of our model. Thus, cells with very low adhesion correspondingly showed low diffusion and a slight increase in adherence resulted in temporary spurts of increased adhesion and cell speed. Finally, we also showed that cell motility is determined by the distribution of adhesion clusters. Numerous small adhesion clusters at the front of a cell result in a higher persistence time than several large adhesion clusters located at the centre of the back of a cell.

   Despite these agreements with experimental data, clear disagreement of the model predictions with experimental observation of course remains. B cells form larger adhesive connections on fibronectin than on collagen IV, and they move faster and more randomly on fibronectin, whereas they move more slowly and more persistently on collagen IV [16]. Interestingly, this observation also disagrees with the universal coupling between cell speed and cell persistence (UCSP) [43] which is reproduced by the Act-model [24] that is at the basis of our model. Briefly, UCSP states that persistence correlates positively with speed. Future

experimental and theoretical work is required for elucidating the causes of this counterexample to the UCSP principle; possibly causes might include polarization cues in the collagen IV matrix facilitating persistent motility, or activation of actin polymerization in response to matrix binding: integrin type $\alpha_5 \beta_1$ signalling induces membrane protrusions, and generates RhoA/Rock-mediated myosin II, whereas $\alpha_v$ integrins signal to reinforce adhesive sites [44].

## Linking cell variability and parameter variability

The experimental studies by Rey-Barroso et al. [16] and Jacobelli et al. [17] showed variability in motility among the cells. Other studies have also described variability among genetically identical *Dictyostelium discoideum* cells in their chemotactic performance [45], and among keratocytes in their shape and speed. Which features of the individual cells underlie such variability in cell motility? By adjusting the adhesion formation rate, adhesion strength and adhesion distribution, we could already capture the different motility modes of B cells on fibronectin and of T cells on casein and ICAM. Still, it would be very interesting to be able to link the changes in these parameters to actual differences between cell populations on different substrates or between individual cells on the same substrate.

An interesting starting point for addressing the variability in lymphocyte motility are measurements of the different subsets of differentiated CD4+ T cells. Th1, Th2, and Th17 subsets have been described to harbour distinct motility properties both *in vitro* and *in vivo*, as well as distinct molecular equipment in terms of adhesion and cytoskeleton dynamics [46, 47]. Th1 and Th17 cells show low speed and displacement, have a low expression of integrin $\alpha_v \beta_3$ and show fluctuating, yet high concentrations of $Ca^{2+}$, whereas Th2 cells have high mobility, high levels of $\alpha_v \beta_3$ and lower and more constant levels of $[Ca^{2+}]$. Interestingly, these differences have been proposed to support distinct search strategies aligning with the fact that these cell subsets target different types of pathogens. The differences in motility and integrin expression correspond well with the observations in our simulations with adhesion-dependent protrusions, where a larger rate of *de novo* adhesion resulted in higher motility as well as more adhesive surface. As such, our model predicts a larger adhesive area for Th2 cells than for Th1/Th17 cells, which can be verified experimentally. By further measurements such as measuring integrin expression levels among individual cells by flow cytometry, or monitoring size and distribution of adhesions with super-resolution microscopy approaches, we can specify our model parameters and assess the predictive value of our model. In general, our study provides a mechanistic framework to identify which processes lead to differential motility and then address this experimentally.

Conversely, experimental measurements can elucidate the current estimates of protrusion and adhesion energies. With the current parameter settings, protrusion energy exceeded the adhesion energy, whereas our literature-based estimates suggest the reverse should be true (S1 Text, [48–51]). However, the referenced studies measured adhesion forces in fibroblasts, osteoclasts and CHO cells, which can have adhesive properties distinct from lymphocytes. For instance, a study on adhesion between a T cells and TNF-$\alpha$ stimulated HUVEC cells measured a de-adhesion work of the entire T cell in the same order as our estimates for protrusion-associated work of a single lattice site [52]. This suggests that our estimated adhesion energies based on other cell types might be largely overestimated. However, how this cell-cell adhesion energy translates to adhesion energies on substrates is yet unclear, and new experimental measurements on lymphocyte adhesion energies on substrates can further improve our model.

## Motility on multiple time scales

An important feature of our model is the possibility to study cell motility at multiple time-scales, especially long-term motility. For long-term behaviour in our model, we mostly observed regular diffusive behaviour or, for more extreme $\lambda_{Act}$ and $p_s$ values, subdiffusive behaviour. This corresponds with the pivoting B cells observed by Rey-Barroso et al. [16]. The other extreme, superdiffusive behaviour, has also been observed in mammalian cells. Harris et al. [11] showed that murine T cells display superdiffusive behaviour, but they have only been tracked for a relatively short time ($\sim 10$ min), so their diffusive behaviour on longer time scales is unknown. In comparison, our simulated cells display persistent, and hence superdiffusive, behaviour at time scales up to 400 to 1000 MCS, which is in the order of minutes, yet they display regular diffusive behaviour at larger timescales. This highlights the existence of multiple time scales in cell motility. Interestingly, the Madin-Darby canine kidney cell in Dieterich et al. [37] have also been tracked for longer time ($\sim 1000$ min) and show three time scales of roughly 0–4 minutes, 4–16 minutes and from 16 minutes onwards, at all of which superdiffusion is observed. Furthermore, cell velocities show long range correlations in time. What causes these long-time correlations and the corresponding long-term superdiffusive behaviour is unclear. Non-trivial behaviour among the factors that determine cell motility might introduce such long-term behaviour, e.g., anomalous rheological properties of the cytoskeleton [53].

An experimental study showed this correlation to be a universal coupling between cell speed and cell persistence (UCSP), mediated by actin flow [43], as actin flow stabilizes cell polarity. In our model, the actin flow is modelled phenomenologically by the Act model [23], which displays this UCSP as well [24]. So the observation that B lymphocytes can display slow persistent motility on fibronectin, and much faster Brownian cell motility seems to disagree with UCSP. Signalling between the cytoskeleton and the matrix may further contribute to the differences in cell motility between the two types of matrices.

## Interplay between cytoskeleton and substrate

In our current model, there is only an explicit interaction between the adhesions and the Act-extension. Specifically, only the protrusion efficiency is directly influenced by the presence of adhesion. However, whether other aspects of the actin network, such as life time and turnover rate of actin networks, are influenced by adhesions is not taken into account.

Where the Act-extension mainly models the actin network at the front of the cell, many locomotion-related processes also involve the contractile components of the cytoskeleton, such as myosin-II. Myosin-II contraction pulls the back of the cell towards the front and can increase cell speed [17]. Within the CPM, myosin-II contractility is modelled indirectly through the perimeter constraint and the contact energy between cell and medium. Changing parameter values for both results in altered speeds within our model (S4 Video). Part of this cortical tension is transferred onto the matrix through adhesions [44, 54]. An interesting question is whether the cortical tension is also influenced by the presence of adhesions. Furthermore, myosin-II is suggested to be a polarization cue and to be transferred to the back of the cell by retrograde actin flow and could possible also alter persistence of cell polarization [43]. An interesting direction for future research would be to study how the retrograde flow is influenced by cell-matrix adhesions and how this may affect the UCSP that also depends on the actin flow.

Next, the retraction of the rear end is slowed down by adhesions if they do not detach. Our current model uses greatly simplified adhesion patch detachment: a stochastic process of loss of sites, and energy requiring retractions of adhesion sites. It ignores the following two

processes: First, myosin-II, besides contracting the rear end, is also involved in the detachment of adhesion patches in T cells [17, 55], as it increases the forces exerted on the adhesions. Second, adhesion detachment at the rear of the cell is also regulated by scaffolding proteins talin and moesin. Both compete for binding sites between integrin and the cytoskeleton, but have different properties. While talin connects the cytoskeleton to integrins, moesin inactivates integrins, thereby decoupling the adhesion from the cytoskeleton at the rear end [56]. Considering the myosin-II dependent detachment and rear-end specific detachment in a next model can further enhance the understanding of the role adhesion cluster size and distribution in cell motility.

So far, we have addressed model improvements regarding intracellular processes. However, cell-matrix interactions are also determined by integrin and matrix properties, including mechanistic feedback between the integrins and matrix. Hence, both matrix rigidity and the cell's ability to generate force influence cell shape and cell motility. When it comes to modelling this feedback, different approaches have been used already in phase-field models of cell motility. In Copos et al. [19], adhesions were modelled as mechanosensitive bonds. In Ziebert et al. [20], adhesions ruptured when they exceeded a maximum length. In Shao et al. [57] the probability of adhesion rupture increased with force. In Löber et al. [21], the matrix deformation was also taken into account, leading to non-trivial motility such as bipedal motility. In Kim et al. [58], a 3D matrix was modelled as individual fibres which a cell can push and pull on. This allows filopodia to sense the matrix stiffness locally, which results in durotaxis.

Modelling matrix deformation or displacement of adhesion sites within the CPM is challenging, but a lot of progress has been made recently. Methods to estimate forces within the CPM cell have been developed, either based on cell shape or on the Hamiltonian [59, 60]. The CPM has also been combined with a finite element method to model matrix traction forces with feedback between the CPM and FEM [59, 61]. Explicit descriptions of cell-matrix adhesions were recently introduced into this framework [62] to describe tension-dependent growth of focal adhesions. In our future work, this methodology will allow us to study the mechanisms by which substrate compliance affects lymphocyte motility.

In conclusion, we propose a simple mechanism by which the ECM can affect the characteristics of lymphocyte trajectories. To this end, we have introduced a novel CPM model that combines the Act model [23] with dynamic cell-matrix adhesions, generating a large repertoire of cell trajectories (Fig 9). We show that a simple model considering actin-driven protrusion formation in interaction with the dynamic formation and detachment of adhesions to the substrate, suffices to reproduce both persistent random walks as well as short-term persistent but long-term subdiffusive random walks. Thus, our simplified model reproduces the motility patterns observed in individual B cells on a fibronectin substrate, such as reduced motility for non-attached cells, walking motility, and pivoting motility due to sustained attachment, as well as the walking and gliding motility of T cells on ICAM or casein substrate. The computational efficiency of the model allowed us to efficiently study both short-term molecular scales as well as the long-term cell behaviour following from it, providing insight into the molecular parameters that explain the plasticity of cell motility due to interaction with substrates. Our study shows that the interplay between adhesion formation, adhesion expansion and adhesion strength may determine the turn-over of the adhesion area which regulates cell speed and persistence.

## Methods

In this work, we model the cells moving on and adhering to flat substrates. The basis of our model is the Cellular Potts model.

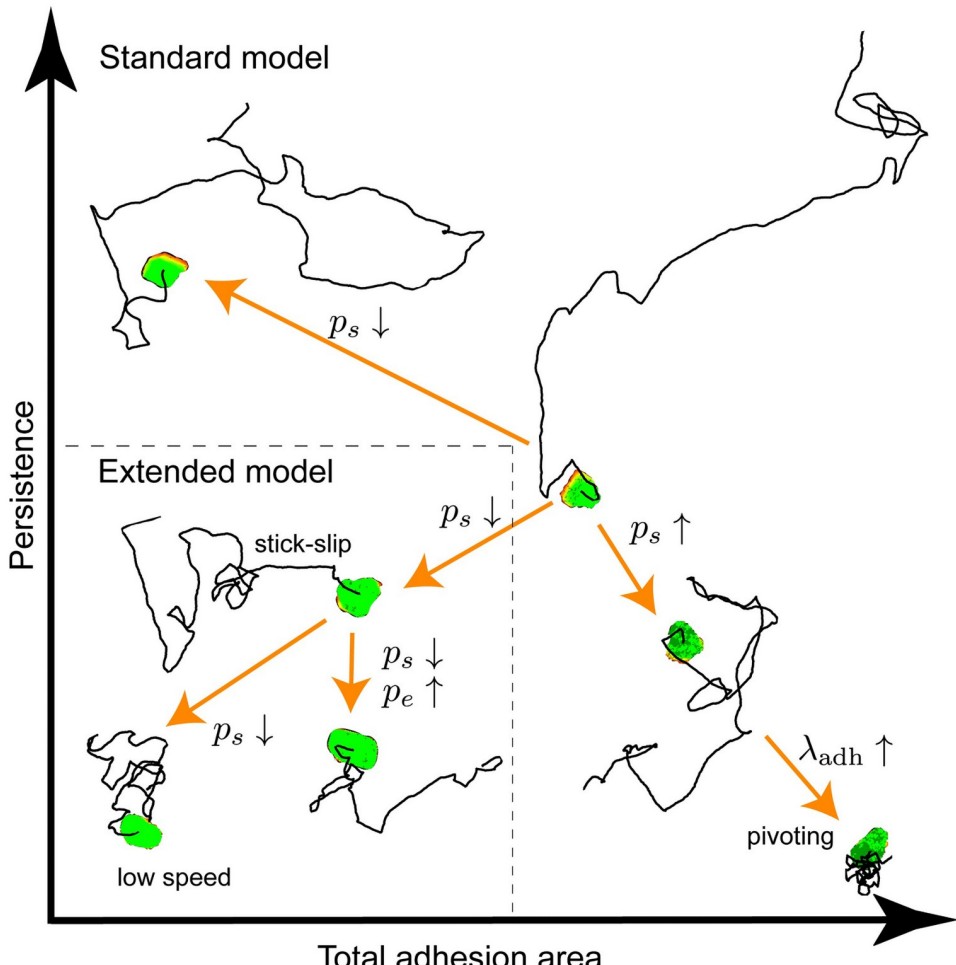

**Fig 9. Overview of the motility modes possible in the model and which parameters govern the transitions between them.** For each motility mode, a representative cell and its trajectory are plotted in a persistence versus total adhesion area plane.

## Cellular Potts model

We use the Cellular Potts model (CPM) to represent a cell on a regular, square lattice. Lattice sites $\vec{x} \in \Lambda \in \mathbb{Z}^2$ are assigned an identity $\sigma(\vec{x})$, with $\sigma(\vec{x}) = 1$ if the site belongs to the cells, and $\sigma(\vec{x}) = 0$ if the cell belongs to the medium. The cell can then be defined as the set $\{\vec{x} \in \Lambda | \sigma(\vec{x}) = 1\}$. The model mimics cell protrusions and retractions through iterative attempts to extend or retract the cell into one of the neighbouring lattice sites, depending on a Hamiltonian function, $\mathcal{H}$, representing passive forces acting on the cell, and a number of active, dissipative processes (e.g., actin protrusion). Together, these represent the balance of forces that drive cell motility. More formally, the algorithm selects a pair of adjacent lattice sites $(\vec{x}, \vec{y})$, where cells are considered adjacent if they are adjacent orthogonal or diagonal neighbours, i.e., $\vec{y} \in \mathrm{NB}(\vec{x}) \setminus \{\vec{x}\}$ where $\mathrm{NB}(\to x)$ is defined as the set of the eight first and second-order neighbours of $\vec{x}$.

The Hamiltonian, $\mathcal{H}$, describes the balance of passive forces produced acting on the cell in terms of a system energy,

$$\mathcal{H} = \sum_{(\vec{x},\vec{y})} J(\sigma(\vec{x}), \sigma(\vec{y}))(1 - \delta(\sigma(\vec{x}), \sigma(\vec{y}))) \; + \lambda_A(A - A_{\text{target}})^2 + \lambda_P(P - P_{\text{target}})^2, \tag{6}$$

The first term describes the contact energies between cell and medium. The second term describes an area constraint, with $A$ and $A_{\text{target}}$ being the area and target area of the cell and, similarly, the third term describes a perimeter constraint with $P$ and $P_{\text{target}}$ the perimeter and target perimeter of the cell.

The total change in energy is given by the change in the Hamiltonian, $\Delta\mathcal{H}$ due to the attempted update in addition to the energy, $\Delta\mathcal{H}_{Act}$ coming from the Act model and $\Delta\mathcal{H}_{Adh}$ giving the energies associated with the detachment of cell-ECM substrate adhesions. The total change in energy $\Delta\mathcal{H}_{\text{total}} = \Delta\mathcal{H} + \Delta\mathcal{H}_{Act} + \Delta\mathcal{H}_{Adh}$ determines the acceptance probability of a copy attempt,

$$P(\Delta\mathcal{H}_{\text{total}}) = \begin{cases} 1 & \text{if } \Delta\mathcal{H}_{\text{total}} < 0 \\ e^{-(\Delta\mathcal{H}_{\text{total}}/T)} & \text{if } \Delta\mathcal{H}_{\text{total}} \geq 0. \end{cases} \tag{7}$$

Here, $T$ controls the amount of random fluctuations in the system. Higher $T$ will allow more thermodynamically unfavourable copy attempts to be accepted. In the Cellular Potts model, time progresses in Monte Carlo step (MCS), which represents a unit of time that allows each lattice site to be updated once on average.

## Cell motility—Act model

Cells move by making protrusions through actin polymerization and form cell extensions like filopodia, pseudopodia, and lamellipodia. Actin polymerization in the CPM has previously been modelled in a phenomenological way in the Act model by Niculescu et al. [23]. This extension adds an extra layer to the CPM, describing the Act values of lattice sites, ranging from 0 to maximum value $\text{Max}_{Act}$. For lattice site $\vec{x}$ newly added to the cell, $Act(\vec{x}) = \text{Max}_{Act}$. At each MCS, the Act values are decreased by 1 until they reach 0. The term $\Delta\mathcal{H}_{Act}(\vec{x} \to \vec{y})$ is subtracted from $\Delta\mathcal{H}$, and can be interpreted as the resulting force from pushing and resistance at the membrane element between $\vec{x}$ and $\vec{y}$. In $\Delta\mathcal{H}_{Act}$, the local geometric mean of Act-values of both the expanding and retracting lattice sites are compared and the lattice site with the highest mean is favoured in the following way:

$$\Delta\mathcal{H}_{Act}(\vec{x} \to \vec{y}) = \frac{\lambda_{Act}}{\text{Max}_{Act}} \left( \left( \prod_{\vec{u} \in V(\vec{x})} Act(\vec{u}) \right)^{1/|V(\vec{x})|} - \left( \prod_{\vec{u} \in V(\vec{y})} Act(\vec{u}) \right)^{1/|V(\vec{y})|} \right), \tag{8}$$

with $V(\vec{x}) = \{\vec{u} \in \text{NB}(\vec{x}) | \sigma(\vec{u}) = \sigma(\vec{x})\}$ describing the neighbourhood of lattice site $\vec{x}$ restricted to the same cell, and $\lambda_{Act}$ is the weight given to this model component.

Adhesion to the matrix is required to fully translate actin polymerization to cell membrane protrusion [40–42], as it transmits the polymerization force to the matrix. We add feedback between the cell adhesions and actin polymerization, by increasing the force produced by polymerization upon increase in adhesion area. This is only up to a threshold adhesion area, after which the protrusion force remains fully activated. We therefore multiply $\lambda_{Act}$ with factor $f$

defined as follows,

$$
f = \begin{cases} b + \dfrac{1 - b}{s} \dfrac{A_{adh}}{A} & \text{if } \dfrac{A_{adh}}{A} \leq s \\[2ex] 1 & \text{if } \dfrac{A_{adh}}{A} > s, \end{cases}
\tag{9}
$$

Here, $A$ denotes the area of the cell, $A_{adh}$ the adhesive area of the cell, $b$ the value of $f$ when there are no adhesions, and $s$ the fractional adhesive area at which $f$ saturates.

## Cell-substrate adhesions

The adhesions of a cell to the extracellular matrix are modelled as a third layer in the CPM. A lattice site $\vec{x}$ in this layer can either contain no adhesion ($Adh(\vec{x}) = 0$), or an adhesion patch ($Adh(\vec{x}) = 1$). Adhesion dynamics are governed by four processes: *de novo* formation of adhesions, adhesion patch expansion, adhesion patch unbinding, and rupture of adhesion through retraction of a cell. We describe each of these processes below.

**New adhesion sites.**   New adhesions form when the cell membrane comes in close enough contact with the extracellular matrix such that integrins can bind to the matrix. This process is dependent on actin polymerization, membrane protrusion and polarized distribution of integrins [27–29]. We model *de novo* adhesion formation through a stochastic process. In each MCS, a grid site within a cell can turn from non-adhesion to an adhesion site with probability $p_s$, if the local geometric mean restricted to the cell of the Act layer exceeds the value 0.75 Max-$_{Act}$, i.e.:

$$
P(\text{new adhesion at } \vec{x}) = \begin{cases} p_s & \text{if } \left( \prod_{y \in V(\vec{x})} Act(\vec{y}) \right)^{\frac{1}{|V(\vec{x})|}} \geq 0.75 \, \text{Max}_{Act}, \\[2ex] 0 & \text{otherwise.} \end{cases}
\tag{10}
$$

**Adhesion patch expansion.**   Once adhesion patches are formed, they can increase in size. Multiple processes underlie this expansion. First, once the cell membrane is attached to the matrix, it fluctuates less, allowing for easier attachment of new integrins [30]. Secondly, the curvature of the cell membrane favours aggregation of integrins [63, 64].

We do not model integrin recruitment and membrane curvature, but choose to model adhesion patch growth phenomenologically. Jacobelli et al. [17] observed that adhesion patches grow radially, with some bias in the direction of the cell front. Hence, we model adhesion patch expansion as an Eden-like growth model [31], known to give roughly circular shapes. While updating the adhesion layer, once a lattice site containing an adhesion is selected to be updated, we also select a random neighbour. If that neighbouring lattice site contains no adhesion, it forms an adhesion with probability $p_e$.

**Adhesion patch unbinding.**   Aside from patch expansion, patch unbinding can also occur, either spontaneously [65] or influenced by myosin-II contraction [17]. Following the observation of concentrical patch detachment [17], an adhesion site $\vec{x}$ in this model can spontaneously detach with a probability depending on the adhesion status of its neighbours.

$$
P(\vec{x} \text{ will unbind}) = p_d \cdot \left( \frac{|\{\vec{u} \in \text{NB}(\vec{x}) \setminus \{\vec{x}\} | Adh(\vec{u}) = 0\}|}{|\{\vec{u} \in \text{NB}(\vec{x}) \setminus \{\vec{x}\}\}|} \right)^2,
\tag{11}
$$

with $\text{NB}(\vec{x})$ the Moore neighbourhood of $\vec{x}$. Thus, the higher the number of non-adherent neighbours, the higher the probability that the site loses its adhesion.

**Adhesion rupture through retraction.**   Adhesions at the cell rear can also unbind by force. Although integrins are known to show catch-slip bond behaviour [66, 67], we simplify the rupture of an adhesion to a constant amount of energy required to break an adhesion upon cell retraction. This determines the term contributing to $\Delta\mathcal{H}_{total}$ $(\vec{x} \rightarrow \vec{y})$:

$$\Delta\mathcal{H}_{Adh}(\vec{x} \rightarrow \vec{y}) = \lambda_{adh} Adh(\vec{y}), \tag{12}$$

with $\sigma(\vec{x}) \neq \sigma(\vec{y})$ and the cell $\sigma(\vec{y})$ retracting.

## Implementation

A measure of time in the CPM is the Monte Carlo Step (MCS). Within one MCS, the expectation is that the $\sigma$ of each lattice site has been updated once. However, many of the proposed neighbouring lattice site pairs share the same $\sigma$ and will thus not result in a changed model state. Therefore, we use a rejection-free algorithm that only considers attempts between neighbours of different $\sigma$ to speed up simulations [68, 69]. Further, the adhesion layer and Act layer of the model are also updated during and after the $\sigma$-update. Act-values and adhesion updates regarding the relocation of the cell are executed immediately during the $\sigma$-update: e.g., for copy attempts that let a cell retract from a lattice site, we do directly update the Act-values and adhesions of that site. After the $\sigma$-update, we update the adhesion layer asynchronously: we iterate, in random order, over the lattice sites within the cell and execute the processes described in the Cell-substrate adhesions subsection. Lastly, we update the Act-layer: every Act-value is diminished by 1 until 0. These three updates together constitute one MCS. The model has been implemented in Tissue Simulation Toolkit and is available in S1 Data.

## Simulation parameters

During our different simulations, many parameter values were kept constant (Table 3). All simulations were done on a 300 × 300 lattice with periodic boundaries with a single cell. Parameter values that were not constant are shown in Table 1. For the simulations in Figs 2, 3 and 4, $p_d$ = 0.0008, and $p_s$ and $\lambda_{adh}$ varied according to the figure legends. For simulations shown in Figs 6 and 7, $p_d$ = 0.001 and again $\lambda_{adh}$ varied according to the figure legends. The Act-only simulations in Figs 3 and 7 were run with all adhesion dynamics parameters equal to zero: i.e., $\lambda_{adh}$, $p_s$, $p_e$, and $p_d$ were all zero. For all simulations, $\lambda_{Act}$ = 240, except for the specific Act-only simulations in Fig 7 with $\lambda_{Act}$ = 120. For the simulations in Fig 8, $p_e$ and $p_s$ were

**Table 3. List of parameter values kept constant during all simulations.** Values are arbitrary units, unless specified otherwise.

| Parameter | Description | Value |
|---|---|---|
| $T$ | temperature | 30 |
| $A_{target}$ | target area | 1000 $px^2$ |
| $\lambda_A$ | weight area constraint | 50 |
| $P_{target}$ | target perimeter | 350 $px$ |
| $\lambda_P$ | weight perimeter constraint | 4 |
| $\lambda_{Act}$ | weight of Act model | 240 |
| $Max_{Act}$ | Act lifetime | 120 $MCS$ |
| $J_{medium,medium}$ | adhesion energy between medium | 0 |
| $J_{cell,medium}$ | adhesion energy between cell and medium | 35 |
| Total MCS | simulation duration | 25000 $MCS$ |

varied, see figure legend. The parameters not mentioned in the figure legend are $p_d = 0.0004$, $\lambda_{adh} = 60$, $b = 0.5$, $s = 0.12$.

## Supporting information

**S1 Fig. Fürth with translational diffusion fits MSD of model better than Fürth without translational diffusion.** Log-log plot of MSD for the four scenarios in Fig 2, similar to Fig 4, with fits of Eqs 1 and 2. Parameters are: A) $\lambda_{adh} = 20$, $p_s = 0.004$, B) $\lambda_{adh} = 100$, $p_s = 0.004$, C) $\lambda_{adh} = 20$, $p_s = 0.02$, D) $\lambda_{adh} = 100$, $p_s = 0.02$.
(PNG)

**S2 Fig. Persistence times obtained from fitting MSD with Fürth with translational diffusion (Eq 2) against adhesion area for different values of $p_s$ and $\lambda_{adh}$.** Parameters are the same as in Fig 3, except that $\lambda_{adh}$ has been limited to 20, 40, 60 because of bad fitting with Eq 2. For reference, the persistence time of the Act model without the adhesion extension is plotted as the black dot.
(PNG)

**S3 Fig. MSD for initial and extended model with fits of Eqs 2 and 4.** Parameters are the same as in Fig 6, with $p_s$ being varied: (A) $p_s = 0.001$, (B) $p_s = 0.004$, and (C) $p_s = 0.0025$.
(PNG)

**S1 Video. Simulations of the model without adhesion-propulsion feedback.** Videos corresponding to Fig 2.
(MP4)

**S2 Video. Simulations of the model with adhesion-propulsion feedback.** Videos corresponding to Fig 6.
(MP4)

**S3 Video. Adhesion growth dynamics influence adhesion cluster size and localisation, cell speed and MSD.** Videos corresponding to Fig 8.
(MP4)

**S4 Video. Effect of perimeter constraint, $\lambda_P$ and cell-medium interfacial energy $J_{cell,medium}$ on cell behaviour.** Parameter values are identical to the blue parameter settings in Fig 8, except where indicated in the video.
(MP4)

**S1 Text. Estimates of parameter units.**
(PDF)

**S1 Data. Model implementation in Tissue Simulation Toolkit.** Also deposited at https://doi.org/10.5281/zenodo.5917626.
(TGZ)

**S2 Data. Interactive Model Implementation using Artistoo.** To use the model, open https://ingewortel.github.io/2021-motility-from-adhesion/. Alternatively, download the Supporting Data File (also deposited at https://dx.doi.org/10.5281/zenodo.5914705), unzip the folder "Artistoo" and open `Artistoo/active-adhesion/index.html` within a web browser.
(TGZ)

## Acknowledgments

G.T. gratefully acknowledges the Indian Institute of Science to serve as Infosys visiting professor at the Centre for Ecological Sciences in Bengaluru. We thank SURFsara for the support and computing time in using the Lisa cluster computer. Martijn de Jong is thanked for his implementation of the rejection-free algorithm of the Cellular Potts model in the Tissue Simulation Toolkit. Babette de Jong is thanked for linguistic advice.

## Author Contributions

**Conceptualization:** Leonie van Steijn, Clément Sire, Loïc Dupré, Guy Theraulaz, Roeland M. H. Merks.

**Data curation:** Roeland M. H. Merks.

**Formal analysis:** Leonie van Steijn.

**Funding acquisition:** Leonie van Steijn, Guy Theraulaz, Roeland M. H. Merks.

**Investigation:** Leonie van Steijn.

**Methodology:** Leonie van Steijn, Roeland M. H. Merks.

**Project administration:** Clément Sire, Loïc Dupré, Guy Theraulaz, Roeland M. H. Merks.

**Software:** Leonie van Steijn, Inge M. N. Wortel, Roeland M. H. Merks.

**Supervision:** Clément Sire, Loïc Dupré, Guy Theraulaz, Roeland M. H. Merks.

**Validation:** Inge M. N. Wortel.

**Visualization:** Leonie van Steijn, Inge M. N. Wortel.

**Writing – original draft:** Leonie van Steijn, Roeland M. H. Merks.

**Writing – review & editing:** Leonie van Steijn, Clément Sire, Loïc Dupré, Guy Theraulaz, Roeland M. H. Merks.

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
