## [Decision Letter · Decision Letter 0]

13 Jul 2021

Dear Merks,

Thank you very much for submitting your manuscript "Computational modelling of cell motility modes emerging from cell-matrix adhesion dynamics" for consideration at PLOS Computational Biology.

As with all papers reviewed by the journal, your manuscript was reviewed by members of the editorial board and by several independent reviewers. In light of the reviews (below this email), we would like to invite the resubmission of a significantly-revised version that takes into account the reviewers' comments.

We cannot make any decision about publication until we have seen the revised manuscript and your response to the reviewers' comments. Your revised manuscript is also likely to be sent to reviewers for further evaluation.

Sincerely,

Alex Mogilner

Guest Editor

PLOS Computational Biology

Douglas Lauffenburger

Deputy Editor

PLOS Computational Biology

Reviewer's Responses to Questions

**Comments to the Authors:**

Reviewer #1: This paper uses a cellular Potts model (CPM) to explore how cell adhesion to surrounding extracellular matrix (ECM) affects the migratory behavior of the cell. The modeling and computing methodology appear to be sound, the exposition of the results is largely clear, and the research seems to have been executed with care. However, a flaw of the presentation is that it does not clearly state the unique contributions of this model vis-à-vis previous modeling work.

The Introduction motivates the work by an open question: what factors cause the different modes of motility? It then argues that these should include both the intracellular actin dynamics and the cell-ECM adhesion. But in the end (L. 92-94), "we are able to reproduce a variety of cell motion types, similar to the behaviour seen in other models that also include those two components [15-17]." This does not represent a novel and significant contribution toward answering the open question.

The Results (pp. 9-20) are clearly presented but contain few surprises. In particular, there is no discussion in this section of how this new model advances our understanding beyond what was known before. Finally, the Discussion section is long and discursive. Besides a summary of the key results, it touches on many things including parameter variability, the time scales, and potential improvements on the current model. But it does not address the issue of novelty and significance of the current work against the backdrop of earlier modeling.

More detailed questions:

Abstract: "With an increasing attachment area and increased adhesion strength, the cells' speed and persistence decreases." This sounds reasonable, and is later born out by Figure 2. But this seems inconsistent with the experimental evidence that B lymphocytes form "larger adhesive connections" on fibronectin than on collagen, but move faster on the former (L. 22-25).

This point is revisited in Discussion (L. 416 onward), but I was unable to understand the explanation or rationalization that follows. Is this a failing of the model? In which aspects? What may be the factors that produced the experimental observations that are unaccounted for?

L. 52-54: does the model explain the apparent contrast between D. Discoideum and T cells cited here? I didn't see where the paper returns to this discrepancy later.

L. 58: "In a second version of the model". At this point, I wonder why 2 versions of the model are presented. Much later on p. 16 (L. 257 onward), we learn that the extended model captures all 3 modes of migration (i - iii on L. 249-251). Then why include the lesser version of the model at all?

L. 108: The subsection on "Modelling cell-matrix adhesions" does not present results and should be in the Methods rather than the Results section.

L. 127: I found the idea of "single adhesion units" intriguing. Is this a single molecular bond? Perhaps this can be explained quickly so the reader need not consult [23].

Caption to Fig. 2: First, the caption should identify the "different motility types": what they are, and to which row they correspond. Second, the 5 vertical dash lines mark the times for the snapshots in the first column. Which corresponds to the snapshots in the second column?

Reviewer #2: In this paper, the authors incorporate the mechanism of cell protrusion and cell-matrix adhesion into Cellupar Potts model and study the relationship between different cell moving modes with those microscopic process including adhesion formation, growth/shrinkage, feedback of adhesion to protrusion, etc. Basically, the questions this paper is focusing on are interesting and Cellular Potts model, as mentioned by the author, is a proper model to study these questions because of the light computational cost and the incorporation of different forces related to cell's movement. But there is one major concern about using 2D Cellular Potts model to study questions related to adhesion and several comments on the way they show their results, which makes the paper less informative and convincing.

1. The main concern about the 2D Cellular Potts Model: one main characteristic of CPM is that the dynamics only happen at the boundary of the cell (where σ(x) is not the same as σ(y)), which is also mentioned in the Methods part of this paper. This means only the adhesion sites at the boundary play a role in the movement of the cell. So, I think it is not proper to relate the cell's movement to the overall adhesion area and adhesion distribution as most of the adhesion sites taken into account in fact do not function/do not enter the ΔH at all. In my opinion, it is better to use a 3D model when the topics are related to the attachment/detachment of the cell to the substrate as then there is a real "surface" that contributes to the adhesion. While now in 2D, it is hard to say whether the interior of the cell contacts with the matrix or not: they cannot exchange sites with the outside but they have adhesion sites. I understand that 3D simulation will greatly increase the computational cost and I do not totally contradict to the use of 2D model but the interpretation should be more careful. The authors need to give reasonable explanations on the interpretation of adhesion with matrix in 2D model and think about how to calculate the "percentage of adhesion area" again. If this explanation is not well given, the fundamental of the model is doubtful.

2. The authors mentions several moving modes of the cells and the three that they are able to obtain with their model are the floating, sliding-stepping and pivoting. The differences they show to discriminate these three modes are the opposite dependence of speed on adhesion area and the sub-diffusive behavior shown in MSD. To my knowledge, these are not the only difference between these modes. For example, there are possibly morphological difference and in stepping mode, there is dynamic attachment/detachment behavior. Of course, it is not fair to require one model to capture all the characteristics but it is necessary for the author to write more clearly what are the main differences of these three modes and which of them can or cannot be captured by the current model. Without this, even I believe the speed and MSD data they show, I do not know to what extent I can relate these three "modes" in the simulation to the three "modes" observed in the experiments. After all, it is possible that not one moving mechanisms show similar speed-adhesion are relation or similar MSD.

3. The right panels of Figure 2 and Figure 6 are not informative enough to support some of their conclusions. It is better to add statistical measurement (e.g. mean and std). For example, in line 181, they mention that the speed fluctuates less in 2E,G than in 2F,H, which is not obvious in the current noisy curves. A way to quantify the fluctuation is needed. And the stick-slip-like behavior mentioned in line 182 and 183 is not obvious, too, considering the fluctuation of instantaneous speed is large and this is only one example among so many simulations.

4. In the section beginning from line 298, the authors mention that they want to study the effect of different distribution of adhesion sites. In line 299, they say that they have mostly looked the effect of adhesions formed at the cell front but in Fig.2, for example, I see lots of adhesion sites in the center and back of the cell. So this sentence is not clear. In Fig.8, they only show the distribution of adhesion cluster size but no statistic information about the sites position (whether mostly at the cell front or other parts) but they conclude in line 325 and 326 that "where those cluster are located" also influences cell motility. What they show in Fig. 8 do not support this conclusion directly. Although I believe it is true because p_s is large, it is better to show the location distribution of the adhesion sites.

5. Still Fig.8, the difference, especially in 8D, is so small. I wonder what prevent the author from choosing another set of parameters that differ from each other more and show larger difference. And in the text, they say that this difference is caused by the distribution (position and size) of the cluster rather than the total adhesion area because the parameters they choose result in the same adhesion area (line 307). But based on their Fig.2 and Fig.6, the adhesion area is a noisy and changing value and thus it is impossible to keep them exactly the same. Then in Fig.8, the difference between the orange and blue is so small. How can they rule out the possibility that this small difference is caused by the small difference in adhesion area?

Reviewer #3: see attached file

**Have the authors made all data and (if applicable) computational code underlying the findings in their manuscript fully available?**

Reviewer #1: **No: **The Supporting Information contains results, plots and movies, but I did not notice codes and data.

Reviewer #2: **No: **No code and raw data are provided

Reviewer #3: Yes

PLOS authors have the option to publish the peer review history of their article (what does this mean?). If published, this will include your full peer review and any attached files.

Reviewer #1: No

Reviewer #2: No

Reviewer #3: No
---

## [Decision Letter · Decision Letter 1]

18 Jan 2022

Dear Prof.dr. Merks,

We are pleased to inform you that your manuscript 'Computational modelling of cell motility modes emerging from cell-matrix adhesion dynamics' has been provisionally accepted for publication in PLOS Computational Biology.

Best regards,

Alex Mogilner

Guest Editor

PLOS Computational Biology

Douglas Lauffenburger

Deputy Editor

PLOS Computational Biology

Reviewer's Responses to Questions

**Comments to the Authors:**

Reviewer #1: I am satisfied with the revision now that the authors have adequately addressed the concerns I raised, and answered the questions. The only objection I have is against the adverb "hopefully" on L. 430 of the revised manuscript, in this sentence: "Future experimental and theoretical work will hopefully elucidate the causes of this counterexample to the UCSP principle".

Reviewer #2: The authors have properly addressed previous concerns.

Reviewer #3: I thank the authors for their detailed, constructive reply to my report. All of my queries have been addressed satisfactorily. I appreciate their explicit discussion of the relation between Markovian full microscopic dynamics and the possibility of non-Markovianity emerging from projections. I was well aware of this possibility but kind of missed the implications within this specific context. But as the authors also point out, nevertheless this does not seem to explain the emergence of superdiffusion in cell migration.

In summary, I am happy with the revised ms. and recommend it now for publication in PLoS Comp. Biol.

**Have the authors made all data and (if applicable) computational code underlying the findings in their manuscript fully available?**

Reviewer #1: Yes

Reviewer #2: Yes

Reviewer #3: Yes

PLOS authors have the option to publish the peer review history of their article (what does this mean?). If published, this will include your full peer review and any attached files.

Reviewer #1: No

Reviewer #2: No

Reviewer #3: No

---

## [Editor Report · Acceptance letter]

9 Feb 2022

PCOMPBIOL-D-21-01092R1 

Computational modelling of cell motility modes emerging from cell-matrix adhesion dynamics

Dear Dr Merks,

I am pleased to inform you that your manuscript has been formally accepted for publication in PLOS Computational Biology. Your manuscript is now with our production department and you will be notified of the publication date in due course.

With kind regards,

Orsolya Voros
